# Topological ultranodal pair states in iron-based superconductors

Chandan Setty[1]*, Shinibali Bhattacharyya [1], Yifu Cao [1], Andreas Kreisel [2] & P.J. Hirschfeld[1]

Bogoliubov Fermi surfaces are contours of zero-energy excitations that are protected in the superconducting state. Here we show that multiband superconductors with dominant spin singlet, intraband pairing of spin-1/2 electrons can undergo a transition to a state with Bogoliubov Fermi surfaces if spin-orbit coupling, interband pairing and time reversal symmetry breaking are also present. These latter effects may be small, but drive the transition to the topological state for appropriate nodal structure of the intra-band pair. Such a state should display nonzero zero-bias density of states and corresponding residual Sommerfeld coefficient as for a disordered nodal superconductor, but occurring even in the pure case. We present a model appropriate for iron-based superconductors where the topological transition associated with creation of a Bogoliubov Fermi surface can be studied. The model gives results that strongly resemble experiments on $FeSe_{1-x}S_x$ across the nematic transition, where this ultranodal behavior may already have been observed.

[1] Department of Physics, University of Florida, 2001 Museum Rd, Gainesville, FL 32611, USA. [2] Institut für Theoretische Physik Universität Leipzig, D-04103 Leipzig, Germany. *email: csetty@ufl.edu

The standard $s_\pm$ paradigm for superconductivity in iron-based superconductors is based on the simple notion that repulsive interband interactions would force a sign change of the order parameter between hole and electron pockets, separated by a near-nesting wave vector at which the magnetic susceptibility is peaked[1,2]. This approach proved rather successful for the iron pnictide superconductors with doping near 6 electrons/Fe, but was questioned in the case of end-point iron based systems where either electron or hole pockets disappear. The latter case includes several FeSe intercalates, including the FeSe/SrTiO$_3$ monolayer system which has the highest $T_c$ of the iron-based class. For these cases, where the standard $s_\pm$ scenario evidently breaks down, a number of exotic alternatives for pairing have been proposed, some of which involve interorbital pair states. Normally such states are energetically disfavored, since they generically force interband pairing of states **k** and $-$**k**, which must occur off the Fermi level and therefore lose the Cooper logarithm that drives robust pairing. However recent works[3,4] have shown that infinitesimal spin-orbit coupling can induce a Cooper log, such that novel interorbital pair states may be expected to occur in special circumstances. Spin-orbit coupling has been proposed to provide the principal source of hybridization between two electron pockets in FeSe monolayers and intercalates, interpolating between $d$-wave and so-called bonding-antibonding $s_\pm$ states[5–7].

At the same time, there has been an ongoing discussion about the possibility of time-reversal symmetry breaking (TRSB) states in the Fe-based materials. These can mix two degenerate representations like $s$ and $d$ at a degeneracy point, leading to a relative $\pi/2$ phase shift as in, e.g., $s + id$ pairing[8,9], or occur in more complicated fashion with arbitrary phase shifts among bands in a system with at least 3 bands[10,11]. Recently, it has been claimed that muSR experiments confirm a predicted TRSB state in highly hole-doped BaFe$_2$As$_2$[12]. The detection of TRSB in bulk FeSe has been reported recently[13] and the sulfur doped compound is under investigation by the same group[14]. Time reversal symmetry can also be broken in the presence of a pseudo-magnetic field arising from interband pairing[15,16]. It has been pointed out that such terms can be generated by spin-orbit coupling in iron based systems[3,4].

Since both spin-orbit effects and time reversal symmetry breaking can individually lead to deviations from the $s_\pm$ paradigm, it is interesting to investigate their interplay, particularly in the context of unusual phenomena observed in the Fe chalcogenides. Recently, a rather unexpected and unusual form of the $T$, $H$-dependent specific heat was observed just after the nematic state disappeared with S doping in the Fe(Se,S) system[17]. The authors of this work were convinced primarily by the change in the magnetic field dependence that the gap structure was changing abruptly at the nematic transition; however, another unusual aspect of the data was the large ($\mathcal{O}(N$ state)) value of the apparent residual $T \to 0$ Sommerfeld coefficient. In a clean superconductor, even with line nodes, $\gamma_s = C/T \to 0$ as $T \to 0$, and while disorder can lead to a nonzero value, STM measurements on these samples suggest that they are very clean, inconsistent with $\gamma_s/\gamma_N$ of $\mathcal{O}(1)$[18]. Both specific heat and STM[18] analyses suggest that the form of the gap function–or at least the density of low-energy quasiparticle excitations—is varying rapidly near the nematic critical point.

In this Article, we propose an explanation for the temperature dependence of the specific heat and the form of the STM conductance spectrum in FeSe$_{1-x}$S$_x$, based upon a prescribed evolution of a superconducting order parameter in the presence of spin orbit coupling that leads to a state with Bogoliubov Fermi surfaces, topologically protected patches of finite area where zero-energy excitations exist in the superconducting state[15]. The only

absolute restriction on the forms of pair wavefunction components arise from the Pauli principle. In a one band system in the absence of spin-orbit coupling, only even parity spin singlet pairing or odd parity spin triplet pairing are allowed. In a multiband system, on the other hand, the additional degrees of freedom allow for novel pairing structures, including odd parity-spin singlet and even parity-spin triplet (both band singlet) components. In order to demonstrate the emergence of a Bogoliubov Fermi surface, we consider a three-pocket model appropriate for iron-based superconductors with a charge and parity ($CP$) symmetric pairing structure. The gap contains interband, TRS and TRSB, spin-triplet-band singlet components. In addition, the pairing structure also contains an intra-band spin singlet-band triplet component with, generally, both isotropic and anisotropic momentum dependences. As a function of the ratio of inter- and intra-orbital pairing, our model exhibits a topological phase transition from a fully gapped $s$-wave superconductor ($\mathbf{Z}_2$ trivial) to a gapless system with extended Bogoliubov surfaces ($\mathbf{Z}_2$ non-trivial). Such a transition is absent when time reversal symmetry is preserved; it is broken here because the interband pairing gives rise to a pseudo-magnetic field.

The topologically non-trivial phase discussed here retains characteristics of both the superconductor and the normal Fermi liquid, e.g., the ratio $C_V/T$ has the usual discontinuity at $T_c$ within mean field theory, but saturates to a non-zero constant as $T \to 0$ due to the finite density of zero-energy excitations on the Bogoliubov Fermi surface[15]. We demonstrate this dichotomy for the specific case of FeSe$_{1-x}$S$_x$ by considering a minimal model for the gaps on the two hole and two electron pockets, qualitatively consistent with the gap structures proposed on the basis of thermodynamic, transport[17] and STM measurements[18]. Within this phenomenology, doping is assumed to tune the anisotropy of the electron pocket gap, driving the topological transition. This simple model can explain essentially all qualitative features observed in FeSe$_{1-x}$S$_x$ across the nematic transition.

## Results

**Model and physical observables.** In the following, we discuss the role of charge-conjugation ($C$), parity ($P$), and time reversal ($T$) symmetries which can all be defined on the momentum space Hamiltonian as

$$-CH(-\mathbf{k})C^{-1} = H(\mathbf{k}) \tag{1}$$

$$PH(-\mathbf{k})P^{-1} = H(\mathbf{k}) \tag{2}$$

$$TH(-\mathbf{k})T^{-1} = H(\mathbf{k}), \tag{3}$$

respectively. While $P$ is a unitary operator represented by a matrix $U_P$, $T$ is anti-unitary and can be denoted as $T = U_T K$ where $U_T$ is a unitary matrix and $K$ complex conjugation (similar notation for $C$ with unitary matrix $U_C$). Before we narrow down our discussions to specific forms of the gap, we begin by highlighting our result for a generic Hamiltonian of the form

$$\hat{H} = \hat{H}_0 + \hat{H}_\Delta$$
$$= \sum_{\mathbf{k}} \Psi_{\mathbf{k}}^\dagger (H_0(\mathbf{k}) + H_\Delta(\mathbf{k})) \Psi_{\mathbf{k}}, \tag{4}$$

where the Nambu operator is defined in the basis $\Psi_{\mathbf{k}}^\dagger = \left( c_{\mathbf{k}i\uparrow}^\dagger, c_{\mathbf{k}i\downarrow}^\dagger, c_{-\mathbf{k}i\uparrow}^\dagger, c_{-\mathbf{k}i\downarrow}^\dagger \right)$ and $c_{\mathbf{k}i\sigma}^\dagger$ is the electron creation operator in pocket $i$ with spin $\sigma$. $H_0(\mathbf{k})$ and $H_\Delta(\mathbf{k})$ are the normal and pairing terms in the Hamiltonian written in momentum space. In the basis chosen above where the orbitals/bands transform trivially under time reversal, the corresponding unitary matrices for $CPT$ symmetries would take the form

$$
\begin{aligned}
U_P &= \pi_0 \otimes \tau_0 \otimes \sigma_0, \\
U_T &= \pi_0 \otimes \tau_0 \otimes i\sigma_y, \\
U_C &= \pi_x \otimes \tau_0 \otimes \sigma_0,
\end{aligned}
\tag{5}
$$

for a two-pocket model. Here, $\pi_i$, $\tau_i$, $\sigma_i$ are Pauli matrices in particle-hole, band (pocket) and spin space, respectively.

We choose the normal state part of the Hamiltonian as $\hat{H}_0 = \sum_{\mathbf{k}i\sigma} \epsilon_i(\mathbf{k}) c^\dagger_{\mathbf{k}i\sigma} c_{\mathbf{k}i\sigma}$ written in the band basis. For the superconducting part, we choose a pairing Hamiltonian having both spin singlet and triplet along with TRS and TRSB components. These terms are written as ($i$ and $j$ are band/pocket indices and $i \neq j$)

$$
\begin{aligned}
\hat{H}_\Delta =&\ \Delta_0 \sum_{i,\mathbf{k}} \left( c^\dagger_{\mathbf{k}i\uparrow} c^\dagger_{-\mathbf{k}j\uparrow} + c^\dagger_{\mathbf{k}i\downarrow} c^\dagger_{-\mathbf{k}j\downarrow} \right) + h.c. - (i \leftrightarrow j) \\
&+ \delta \sum_{i,\mathbf{k}} \left( c^\dagger_{\mathbf{k}i\uparrow} c^\dagger_{-\mathbf{k}j\uparrow} - c^\dagger_{\mathbf{k}i\downarrow} c^\dagger_{-\mathbf{k}j\downarrow} \right) + h.c. - (i \leftrightarrow j) \\
&+ \sum_{i,\mathbf{k}} \Delta_i(\mathbf{k}) \left( c^\dagger_{\mathbf{k}i\uparrow} c^\dagger_{-\mathbf{k}i\downarrow} - c^\dagger_{\mathbf{k}i\downarrow} c^\dagger_{-\mathbf{k}i\uparrow} \right) + h.c.
\end{aligned}
\tag{6}
$$

Here $\Delta_0$ and $\Delta_i(\mathbf{k})$ are the TRS components of the pairing and $\delta$ is the degree of TRS breaking; $\Delta_i(\mathbf{k})(\Delta_0)$ appears as an intraband (interband) spin-singlet (spin-triplet) term. It can be verified that the pairing Hamiltonian above, along with the band dispersion, is $CP$ symmetric. Therefore, following the arguments of Agterberg et al.[15], one can similarity transform the Hamiltonian to a basis where it is completely anti-symmetric; hence, the Pfaffian of such a system is well defined.

Substituting the pairing structure $\hat{H}_\Delta$ in the total Hamiltonian $\hat{H}$ and evaluating the determinant, one finds that the Pfaffian is given by

$$
\begin{aligned}
\mathrm{Pf}(\Delta_0, \Delta_i, \delta, \epsilon_i) =&\ \Delta_0^4 + \Delta_1^2(\mathbf{k})\Delta_2^2(\mathbf{k}) + \left( \delta^2 + \epsilon_1(\mathbf{k})\epsilon_2(\mathbf{k}) \right)^2 \\
&- 2\Delta_0^2 \left( \delta^2 - \epsilon_1(\mathbf{k})\epsilon_2(\mathbf{k}) \right) \\
&+ \Delta_1(\mathbf{k})^2 \epsilon_2(\mathbf{k})^2 + \Delta_2(\mathbf{k})^2 \epsilon_1(\mathbf{k})^2 \\
&+ 2(\delta^2 - \Delta_0^2) \Delta_1(\mathbf{k})\Delta_2(\mathbf{k})
\end{aligned}
\tag{7}
$$

We next determine the condition for the existence of a Bogoliubov surface by checking for a change in sign of Pfaffian. The function $\mathrm{Pf}(\Delta_0, \Delta_i, \delta, \epsilon_i)$ acquires arbitrarily large positive values for arbitrarily large (in magnitude) dispersions $|\epsilon_i(\mathbf{k})|$. To determine if the Pfaffian turns negative, we minimize $\mathrm{Pf}(\Delta_0, \Delta_i, \delta, \epsilon_i)$ with respect to $\epsilon_i$ ($\epsilon_1(\mathbf{k}) \neq \epsilon_2(\mathbf{k})$). For a non-zero $\delta$ and $\Delta_0$, the minimum value depends on the relative sign of $\Delta_1(\mathbf{k})$ and $\Delta_2(\mathbf{k})$ and is given by

$$
\mathrm{Min}[\mathrm{Pf}] = \begin{cases} \delta^2 \left( |\Delta_1(\mathbf{k})||\Delta_2(\mathbf{k})| - \Delta_0^2 \right) & \text{if } \Delta_1(\mathbf{k})\Delta_2(\mathbf{k}) > 0 \\ \Delta_0^2 \left( |\Delta_1(\mathbf{k})||\Delta_2(\mathbf{k})| - \delta^2 \right) & \text{if } \Delta_1(\mathbf{k})\Delta_2(\mathbf{k}) < 0. \end{cases}
\tag{8}
$$

Given that the Pfaffian is large and positive for momenta $\mathbf{k}$ corresponding to energies far from the Fermi level, it changes sign only if the minimum is negative. Looking at Eq. (8) for the case when the order parameters on both the bands have the same sign ($\Delta_1(\mathbf{k})\Delta_2(\mathbf{k}) > 0$), this can be achieved if the following two conditions are fulfilled. First, there must be a nonzero time-reversal symmetry breaking component, $|\delta| > 0$. Second, the term in parentheses has to become negative, leading to the condition $|\Delta_0|^2 > |\Delta_1(\mathbf{k})||\Delta_2(\mathbf{k})|$, i.e., the magnitude square of the inter-band pairing exceeds that of the product of the intra-band pairings on the two pockets. On the other hand, when the order parameters on the two bands have opposite signs, the roles of $\delta$ and $\Delta_0$ are switched, as can be deduced from Eq. (7) for the Pfaffian. In the numerical calculations to follow, we have set $\delta = \Delta_0$ so that the results are independent of the relative sign of

the two gaps. With the above conditions satisfied, a Bogoliubov surface emerges from a fully gapped superconducting phase giving rise to a topological phase transition at a critical combination of the order parameters. At low temperatures far away from $T_c$, the low energy excitations close to the Bogoliubov surface resemble that of a normal metal yielding a finite residual specific heat. Hence, the system represents a unique example where one can have non-zero superconducting order parameter coexisting with a fully gapless Fermi surface. In contrast to the example of gapless super-conductivity in disordered superconductors[19], this phase can occur in a completely clean system, and the Bogoliubov surface does not coincide in momentum space with the normal metal Fermi surface. Such a state is characterized by a specific heat jump at the superconducting critical temperature, but a non-zero Fermi level density of states and Sommerfeld coefficient $C_V/T$ as $T \rightarrow 0$. We therefore adopt the name ultranodal superconducting state to indicate that the phase space for Bogoliubov quasiparticle excitations is larger than in either the point or line nodal case familiar from studies of unconventional superconductivity.

**Case of Fe(Se,S).** As a concrete example to highlight the physics discussed above, we consider a simplified three-pocket model ($i = X, Y, \Gamma$) in two dimensions to capture the essential electronic structure of iron-based superconductors, and in particular the FeSe$_{1-x}$S$_x$ system that has been noted to display anomalous thermodynamics for doping beyond the nematic transition[17]. We set the band dispersions to be quadratic, centered around $\Gamma$ and $X/Y$ points such that the quantities that influence the existence and location of the Bogoliubov Fermi surfaces are the gap functions. Choosing an intra-pocket pairing ansatz of the form $\Delta_j(\mathbf{k}) = \Delta_{ja}(\mathbf{k}) + \Delta_j$, where $\Delta_{ja}(\mathbf{k})(\Delta_j)$ is the anisotropic (isotropic) component on pocket $j = X, Y, \Gamma$, we decrease with doping the isotropic components on each pocket for a fixed inter-pocket pairing $\Delta_0$ and anisotropic intra-band component $\Delta_{ja}$, as sketched roughly in Fig. 1. This is in accordance with the data and conclusions in ref. [17] for FeSe$_{1-x}$S$_x$ where, as a function of sulfur doping, all the pockets were posited to become nodal or close to nodal across the nematic quantum critical point. The anisotropic component on each pocket, $\Delta_{ja}(\mathbf{k}) = \Delta_{ja}(k_x^2 - k_y^2)$, with $\mathbf{k}$ measured from the center of

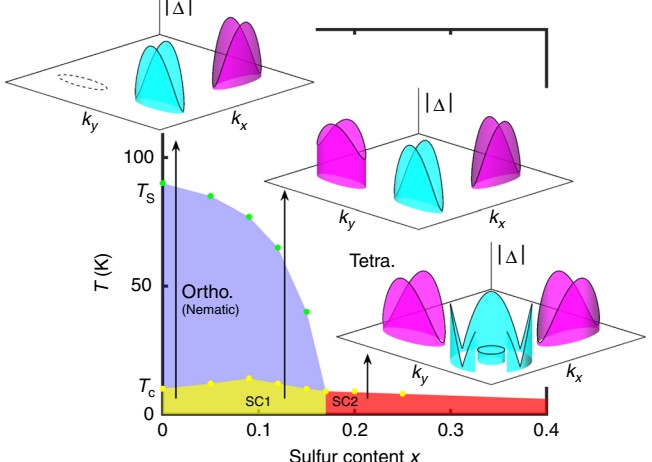

**Fig. 1 Schematic plot of the phase diagram and intra-pocket order parameters used on the electron and hole bands in different phases across the transition.** Light blue and purple colors refer to assumed opposite signs of order parameters on hole and electron pockets, which are, however, qualitatively irrelevant for the conclusions discussed here. Dashed Fermi surface in $x = 0$ case refers to pocket that has not been observed spectroscopically. Data from refs. [17-24].

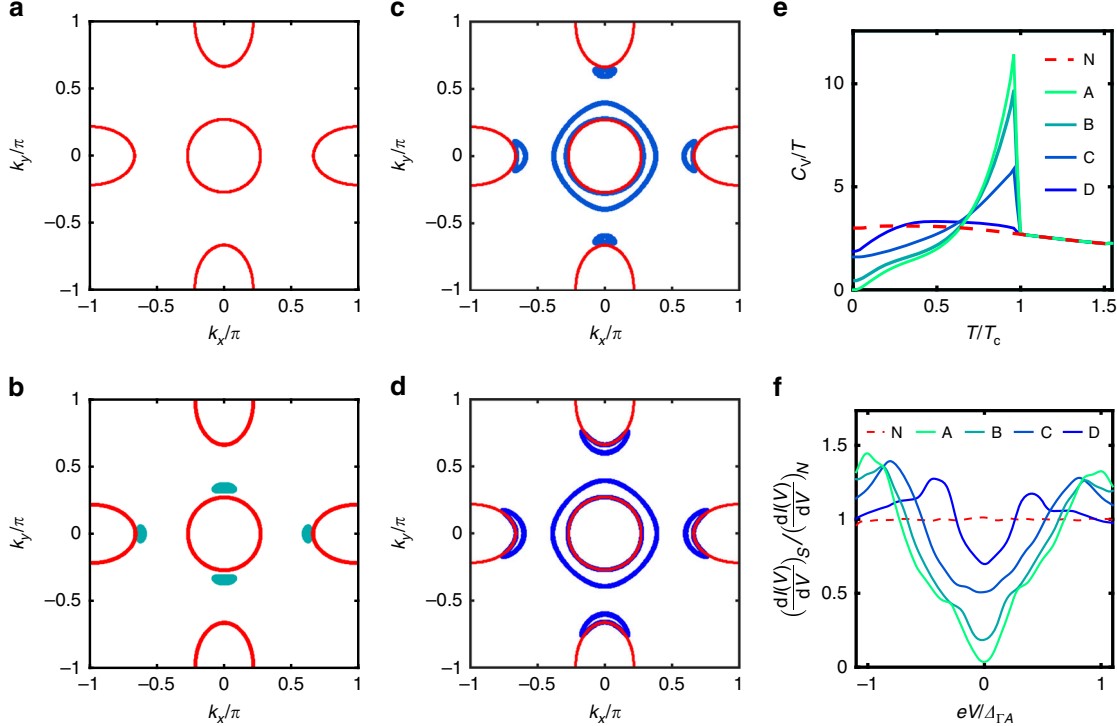

**Fig. 2 Transition into the Ultranodal state. a–d** Normal state Fermi surface (red contour) and Bogoliubov Fermi surface in superconducting state (blue/green patches) for different values of the isotropic gap parameters on each pocket. Note that while results are plotted over a putative 1st Brillouin zone, the model is actually continuous. The inter-band gap component is chosen to be $\Delta_0 = 0.4$ and the time reversal broken component $\delta = \Delta_0$. Anisotropic gap components are $\Delta_{\Gamma a} = 0.1$, and $\Delta_{Xa} = \Delta_{Ya} = 0.4$, Isotropic gap components are given as $[\Delta_\Gamma, \Delta_X, \Delta_Y]$ in in each of the set-A: [0.40, 0.35, 0.35], B: [0.35, 0.27, 0.35], C:[0.16, 0.20, 0.25], D:[0.07, 0.07, 0.07]. Note the $C_2$ symmetry of the nodes for larger isotropic gaps, consistent with ARPES. **e** Temperature dependence of the specific heat $C_V/T$ for the sets of gap components on each pocket (A–D). **f** Tunneling conductance $dI/dV$ normalized to normal state value vs. STM bias $eV$, normalized to hole pocket intraband gap $\Delta_{\Gamma A}$ evaluated at temperature $T = 0.07 T_c$. Curves are calculated by convolving density of states $\rho(E)$ of 3-pocket model with Fermi function derivative[25]. The sets of gap values A–D span the nematic transition, with decreasing isotropic gap component $\Delta_{ji}$ between gapped/near nodal state (A) and ultranodal states (B–D). Normal state conductance (red) is also given for reference.

the pocket, is chosen to yield $C_2$-symmetric gap structures in the nematic superconducting phase consistent with both ARPES[20] and STM[21]. Note the gap structures discussed here are simply plausible guesses respecting the symmetry of the various phases and agreeing qualitatively with the evolution suggested by experiment; as yet, there is no microscopic theory of the ultranodal state.

As argued above, when the isotropic intraband component becomes sufficiently small in the presence of interband pairing and spin orbit coupling, the Pfaffian changes sign, and the spectrum of low-energy excitations is altered dramatically. As shown in Fig. 2a–d, as the ratio of intraband isotropic to anisotropic components decreases with doping, the system develops a Bogoliubov Fermi surface, which grows and becomes more $C_4$ symmetric as nematic order disappears. We emphasize that it is impossible within the present model framework to associate a unique set of gaps with a particular doping. However, one can plot the specific heat, which allows a rough comparison with experiment. Figure 2e shows a plot of $C_V/T$ as a function of temperature for the same sets (A–D) of isotropic gap components on the individual pockets. $C_V/T$ goes to zero as $T \to 0$ for larger isotropic components (A), but saturates at a nonzero value for smaller ones (B–D), reflecting the ultranodal state. Hence a system such as FeSe$_{1-x}$S$_x$ can exhibit properties of both a superconductor and a normal Fermi liquid. Close to critical values of the isotropic gap (see Fig. 2b), the Bogoliubov surface shrinks continuously to a point.

This behavior is reminiscent of experiments on FeSe$_{1-x}$S$_x$[17]: as one dopes through the nematic transition, the specific heat ratio $\Delta C_V/C_N$ at the transition is observed to fall, while at the same time

the residual Sommerfeld coefficient in the superconducting state increases. Thermal conductivity[17] exhibits a similar low temperature evolution. The abruptness of the transition is not entirely clear from the thermodynamic data, since the samples are spaced relatively far apart in doping. A recent STM experiment[18] employed a more closely spaced sequence of S-dopings, and was able to establish that a transition took place between $x = 0.13$ and $x = 0.17$, corresponding roughly to the nematic transition. At this transition, the coherence peak position in the conductance spectrum red-shifted weakly, while the zero bias conductance became nonzero, growing with increased S-doping.

Precisely this behavior is seen in the superconducting density of states across the topological transition into the ultranodal state, as shown in Fig. 2f. As expected, with the development of the ultranodal state, the zero-energy value of the DOS increases due to the presence of Bogoliubov Fermi surfaces of increasing size. In addition, the weight in the coherence peaks is suppressed and their position weakly red shifted in the ultranodal phase, resulting in a gap-filling rather than a gap-closing phenonmenon very similar to experiment.

## Discussion

It is important to reemphasize the distinction between the two ways in which pairing terms can break TRS—the time reversal operator $T = U_T K$; hence, TRSB can occur either because the order parameter acquires an imaginary component (type 1) or/and because the spin sector in the pairing term does not transform properly under the unitary part of the time reversal operator

(type 2). From our analysis it is clear that a pure intra-pocket type 1 TRSB cannot yield a Bogoliubov surface since the Pfaffian preserves sign (irrespective of the number of input bands). Therefore, one necessarily requires a non-zero (even infinitesimally small, as is evident from Eq. (8)) type 2 TRSB, $\delta^2 > 0$, to achieve a non-trivial phase when the sign of the order parameter is the same on the two pockets.

Theories with pure interpocket pairs are thought to be generally unstable as they can exhibit negative superfluid density[22,23], so it behooves us to check that such an instability does not occur in the present case. In the supplementary material (See Supplementary Fig. 6), we present a calculation in various limits, whose main conclusion is that intrapocket gaps push the superfluid density to remain positive. Since the physical situation we describe here includes intrapocket gaps larger than interpocket ones, our theory is indeed stable. The general situation is interesting, but we postpone a fuller discussion to a subsequent publication.

Finally, we note that combined thermodynamic and ARPES data, together with our analysis, point to deep minima on at least one of the $\Gamma$ hole pockets coinciding with those on the electron pockets at the same momentum direction near the nematic quantum critical point. The general condition for the Pfaffian to change sign requires that the product of two intraband gaps be smaller than the TRS interband gap (TRSB component $\delta$) when two pockets have the same (opposite) sign of the order parameter. Since interband gaps and TRSB breaking components are generically small, this requires that nodal or deep minima align such that the product of intraband hole and electron gaps $\Delta_e(\mathbf{k})\Delta_h(\mathbf{k})$ is also small. Fig. 1 shows how such a coincidence of nodal structures along the $k_x$ and $k_y$ axis might occur in the tetragonal phase. We note also that intrapocket nodal structures are essential to allow for the formation of the ultranodal state without the need for large intraband interactions, typically required to overcome the missing Cooper logarithm due to pairing states away from the Fermi level.

In summary, we explored the possible existence of an ultranodal superconductor—a state of matter uniquely characterized by topologically protected extended zero-energy surfaces (Bogoliubov surfaces)—in multi-band superconductors with spin-1/2 pairs such as iron-based systems. This phenomenon occurs in the presence of a spin-orbit coupling-induced triplet pairing component, interband pairing, and type-2 broken time reversal symmetry. We derived conditions for the existence of such surfaces and studied the behavior of the specific heat, the Sommerfeld ratio $C_V/T$, and the density of states close to and away from the transition. We argued that the topologically non-trivial phase retains thermodynamic and electronic properties of both the superconducting state and the normal Fermi liquid. Finally, we argued that our results have direct, immediate experimental relevance by examining recent evidence for an excess of low energy quasiparticle states in the clean iron-based superconductor $\mathrm{FeSe}_{1-x}\mathrm{S}_x$ near its nematic transition. We showed that theoretical specific heat and conductance spectra agreed remarkably well with experiment across the transition, and concluded that an ultranodal state can exist in this system.

## Methods

**Model details**. The calculations were performed by taking simple parabolic dispersion for the electronic structure in continuum space. Each of the pockets were chosen to have a quadratic dispersion, specifically

$$\epsilon_\Gamma(\mathbf{k}) = -\frac{4\alpha}{\pi^2}\mathbf{k}^2 + E_+$$

$$\epsilon_X(\mathbf{k}) = \frac{4\alpha}{\pi^2}\left[\left(\frac{k_x - \pi}{1+\epsilon}\right)^2 + \left(\frac{k_y}{1-\epsilon}\right)^2\right] - E_-$$

$$\epsilon_Y(\mathbf{k}) = \frac{4\alpha}{\pi^2}\left[\left(\frac{k_x}{1-\epsilon}\right)^2 + \left(\frac{k_y - \pi}{1+\epsilon}\right)^2\right] - E_-$$

with the parameters $\alpha = 2$ and $E_+ = 0.6$, $E_- = 0.6$, $\epsilon = 0.2$ and additonally inserting symmetry related electron bands having band minima at $(0, -\pi)$ and $(-\pi, 0)$. In the numerical implementation, arbitrary energy units are chosen which then are assigned to the real units by fixing the critical temperature $T_c$ and the position of the coherence peaks $\Delta_{\Gamma A}$, see Fig. 2e, f. The order parameters are explicitly given by

$$\Delta_\Gamma(\mathbf{k}) = \Delta_\Gamma + \frac{4\Delta_{\Gamma a}}{\pi^2}(k_x^2 - k_y^2)$$

$$\Delta_X(\mathbf{k}) = \Delta_X + \frac{4\Delta_{Xa}}{\pi^2}\left[-\left(\frac{k_x - \pi}{1+\epsilon}\right)^2 + \left(\frac{k_y}{1-\epsilon}\right)^2\right]$$

$$\Delta_Y(\mathbf{k}) = \Delta_Y + \frac{4\Delta_{Ya}}{\pi^2}\left[\left(\frac{k_x}{1-\epsilon}\right)^2 - \left(\frac{k_y - \pi}{1+\epsilon}\right)^2\right],$$

and the corresponding order parameters on the symmetry related electron bands. A momentum grid of size $200 \times 200$ was used for the calculation of specific heat. Integrations were performed from $-2\pi \le k_{x,y} \le 2\pi$. All the required gap components have been mentioned in the main text under Fig. 2. The calculation of LDOS was carried out on a momentum grid of size $800 \times 800$ points and on a frequency grid of 300 points. Artificial broadening was set to 0.004.

To obtain specific heat, one can start from calculating the entropy $S$ of the free Fermi gas of Bogoliubov quasiparticles, in terms of the spectrum $E_\mathbf{k}$ of the Hamiltonian $H(\mathbf{k})$ and use $C_V = VTdS/dT$ to obtain

$$C_V = 2\sum_\mathbf{k} \frac{-\partial n(E_\mathbf{k})}{\partial E_\mathbf{k}}\left(\frac{E_\mathbf{k}^2}{T} - E_\mathbf{k}\frac{\partial E_\mathbf{k}}{\partial T}\right),$$

where $n(x) = 1/(\exp(x/T) + 1)$ is the Fermi function and the temperature dependence of the order parameter was assumed to follow a mean field behavior.

## Data availability

All data generated or analyzed during this study are included in this published article (and its supplementary information files).

## Code availability

All codes used to generate or analyze the results of this study are available from the corresponding author on reasonable request.

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

## Acknowledgements

We thank D. Agterberg and Y. Wang for useful discussions. This research was supported by the Department of Energy under Grant No. DE-FG02-05ER46236.

## Author contributions

C.S. and P.H. conceived the current idea, and C.S. carried out the analytical calculations in a simple model. S.B. carried out subsequent analytical calculations and most numerical calculations under the supervision of P.H. and C.S. A.K. participated in initial discussions framing the problem, and performed numerical calculations of multiorbital spin fluctuation pairing-driven $d$-wave state to rule out conventional explanation. Y.C. performed superfluid density calculations under the supervision of P.H. and C.S. All authors contributed to the analysis of the results and participated in the writing of the paper.

## Competing interests

The authors declare no competing interests.
