## [Peer Review File · Nature Communications]

Reviewers' comments:

Reviewer #1 (Remarks to the Author):

The reviewed article "Topological Ultranodal pair states in iron-based superconductors" by Setty et al. proposes an exotic new pairing state near the nematic transition in the iron selenides. This is a time-reversal symmetry-breaking (TRSB) pairing state with interband pairing, which leads to the formation of Bogoliubov Fermi surfaces where point nodes are expected. This results in low-temperature thermodynamic behaviour closer to a Fermi liquid as opposed to a nodal superconductor, which the authors call "ultranodal". This scenario can explain the abrupt increase in zero-energy spectral weight observed upon doping FeSe as reported in references 15 and 16.

The manuscript is thought-provoking. However, I don't think that the manuscript warrants publication in Nature Communications. The main novelty here is the application to the pnictides, but technically the authors don't seem to significantly go beyond the analysis of refs. 13 and 14. More interesting questions are left unaddressed: how could such a TRSB state arise? Considering the large residual density of states, is such a superconducting phase stable?

Moreover, I found the manuscript to be a bit sloppy. Specifically:

1. The authors introduce the theory for a two-band model, but then immediately switch to a three-band model, thus rendering the Pauli matrix notation of the introduction confusing.
2. The meaning of fig. 1 is unclear: do the insets show the form of the gap used in their simulations? If so, where do the Bogoliubov Fermi surfaces appear? Also, what is the new pocket appearing in the $x=0.22$ sketch?
3. I don't think that the term "ultranodal" is helpful. The low temperature behaviour of the heat capacity is clear from the term "Bogoliubov Fermi surface".
4. Is plot fig 2e supposed to be the density of states or the tunneling conductance? The former terminology is used in the text, whereas the latter is used for the figure. It seems to me that a calculation of the density of states would be simpler and much less noisy.
5. The authors appear to mischaracterize refs. 13 and 14 when they state that "Time reversal symmetry can also be broken when spin-orbit coupling generates a pseudo-magnetic field differentiating between up and down spins." The spin-orbit coupling in refs. 13 and 14 is not essential for the TRSB, nor for the appearance of Bogoliubov Fermi surfaces.

Reviewer #2 (Remarks to the Author):

The authors considered a phenomenological model for superconductivity in FeSeS, which is shown in experiments to have a finite ratio of the Sommerfeld coefficient in specific heat in the superconducting state, implying the existence of gapless excitations. In the theoretical model, intra- and inter-pocket pairings, singlet and triplet, and time-reversal symmetry breaking, are included in such a way that the BdG quasiparticles have a Fermi surface, and the superconducting state is dubbed ultranodal. This naturally explains the behavior of the puzzling specific heat. The authors believed that the transition from fully gapped state to the ultranodal state is topological in nature, as reflected in the change of the sign of the Pfaffian of the matrix Hamiltonian.

While the proposal sounds interesting, I have some comments for the authors to consider before I could make a decision on the publication of the manuscript.

[1] The ultranodal state requires fine tuning of the parameters, and as mentioned in the text, in some cases, the transition is even sensitive to some infinitesimal tuning of inter-pocket coupling. The need of fine tuning of the various pairing parameters, and the lack of adiabatic continuity in the inter-pocket coupling, make the theory less compelling.

[2] The need of time-reversal symmetry breaking is also a concern. Is there any evidence of this symmetry breaking in experiments?

[3] The Sommerfeld coefficient in experiments indicate, at most, the existence of low lying quasiparticles. But the model here implies more than that. For example, it actually implies the existence of dispersive low lying quasiparticles, and this should imply the existence of such states in transport, e.g., thermal conductivity. Is there any evidence in this respect?

[4] The key of the phenomenological theory is the multiorbital physics that enables the spin-singlet and spin-triplet to be both even and odd under inversion, since the orbitals can make up the antisymmetry. In this respect I would like to draw the authors' attention to the earliest paper on possible p-wave spin-singlet orbital-triplet pairing scenario for iron pnictides, Phys. Rev. Lett., 101 (2008) 057008. Without recognition, that state also has a BdG Fermi surface.

[5] As the authors mentioned, the inter-pocket pairing does not enjoy the infrared divergence in the pairing susceptibility. It is here that the authors argued that an infinitesimal inter-pocket coupling would rescue the logarithmic infrared divergence. While this is indeed correct because pocket hybridization enables the contents of the Bloch state to redistribute, adiabatic continuity implies that such a divergence must be proportional to the forth power of the infinitesimal coupling. It is unclear to me why such a weak effect should be able to drive superconductivity and be more favorable than intrapocket pairing. Without a mechanism to stabilize the model under concern, the theory is purely phenomenological, and is not exclusive, since other phenomenological fitting would work as well.

[6] Last and most importantly, given the existence of the BdG Fermi surface and hence a finite density of propagating zero-energy BdG quasiparticles (which now carry charge since they are no longer symmetric combinations of particles and holes), the authors should check the stability of the superconducting state to phase twisting, namely, the superfluid density. It is known, at least in one example, that the existence of the BdG Fermi surface leads to zero or even negative superfluid density. See EPL, 85 (2009) 57007.

To summarize, the paper in the present form lacks the scientific rigor for publication in NC.

Reviewer #3 (Remarks to the Author):

In this manuscript, the authors propose an interesting superconducting state of $\text{FeSe}_{1-x}\text{S}_x$ in the tetragonal phase ($x > x_c = 0.17$). Recent STM study [15] reported that the superconducting gap structure becomes suddenly isotropic for $x > x_c$, and the residual DOS increases with x . This experiment attracts great attention because it indicates the intimate relation between superconductivity and nematic order. To explain nontrivial superconducting DOS of $\text{FeSe}_{1-x}\text{S}_x$ for

$x > x_c$, the present authors applied the idea of superconductivity with Bogoliubov Fermi surface, which appears when the time-reversal symmetry (TRS) breaks, and the inter-band Cooper pairs dominate over the intra-band ones. The obtained DOS in Fig.4 seems to explain the result of STM study in Ref. [15].

I think that the exotic superconductivity proposed in this manuscript is very interesting, so the present study will stimulate many experimentalists in this field. On the other hand, the authors do not discuss possible origin of this TRS breaking inter-band pairing state. (The idea of superconductivity with Bogoliubov Fermi surface was first proposed in Ref. [13].) Actually, one has to assume very large inter-band pairing interaction for $x > x_c$, much larger than the intra-band pairing interaction for $x < x_c$. However, this compound becomes weak-coupling with increasing x . To explain monotonous increase of residual DOS with $x (> x_c)$, the inter-band pairing interaction should remain large away from the nematic critical point $x = x_c$.

The Bogoliubov Fermi surface (shown by blue lines in Fig.2) appears at the momentum k , in the case that two bands with $E_1(k)$ and $E_2(k)$ are very close to the Fermi level. For this reason, the band structure near the Fermi level is very important information. Therefore, I believe that the authors should show the model band structure, and make comparison with experimental band structure in (for instance) Phys. Rev. B 96, 121103 (2017) in detail.

In summary, the proposed exotic superconductivity with Bogoliubov Fermi surface for $\text{FeSe}_{1-x}\text{S}_x$ is very interesting. The present will promote many experimental studies to detect the Bogoliubov Fermi surface and to find the evidence of the TRS breaking in the superconducting state. On the other hand, the present authors do not discuss possible origin of the inter-band pairing interaction based on the characteristic band structure of $\text{FeSe}_{1-x}\text{S}_x$ near E_F . The idea of Bogoliubov Fermi surface was originally discussed in Ref. [13] in a general model. An alternative explanation for the residual DOS, which increases with x monotonically, might be given by dirty d-wave superconductivity. Therefore, I feel a hesitation to recommend the present manuscript for publication in Nature Communications.

Response to Referees:

General response 1 to all referees: time reversal symmetry breaking. In our work, we propose a model that implies an explicit time reversal symmetry breaking in the superconducting state. For the specific material FeSe, μ SR measurements have been done very recently. Indeed, these (unpublished) experimental investigations in the parent compound of FeSe_{1-x}S_x observe a significant TRSB signal in the superconducting state (presented by Kohei Matsuura from U. Tokyo, APS March meeting 2019 <https://meetings.aps.org/Meeting/MAR19/Session/E10.6>). More measurements in the over-doped FeSeS are being performed currently and should be available soon <https://meetings.aps.org/Meeting/MAR19/Session/E10.7>. We note that the TRSB order parameter in our work is of “Type 2” where spin-up and spin-down triplets have different pairing amplitudes. Such non-unitary pair states and their corresponding mechanisms are scarcely studied in literature, and we are unaware of any theoretical proposals in the context of high T_c superconductors (cf. Weng et al, Phys. Rev. Lett. **117**, 027001 (2016) for a relevant low- T_c example). Therefore, our prediction for the existence of such exotic states has the potential to open-up a flurry of experimental as well as theoretical activity. We believe that the current lack of theoretical understanding of these very unusual experimental results calls on the community to entertain such non-unitary TRSB pairing states.

General response 2 to all referees: novelty. With regard to the novelty of our investigations, we wish to state at the outset that the theoretical tools employed in our manuscript are not new, nor did we claim in the manuscript that they were. The concept of a Bogoliubov quasiparticle Fermi surface was, to our knowledge, first proposed by Berg et al., PRL 100, 027003 (2008), and further developed by Wang and Vafeek, Phys. Ref. B bf88, 024506 (2013). The core ideas involving *topologically protected Bogoliubov surfaces* were first introduced by Agterberg, Brydon and Timm in the context of spin $j = 3/2$ fermions, followed by a more comprehensive PRB study by the same authors along with Schnyder. Developing the theoretical formalism for a novel topological phase of matter is not our intention nor does it form the central claim of our work. The purpose of our paper is to point out a possible first experimental realization of these exotic topological pairing states proposed in the literature, and a route to creating them that may be implicit in the work of Agterberg et al, but was never explored. Our central claim is that such states of matter may have already been observed in traditional, spin-1/2 paired, apparently non-topological contexts without having been recognized. Pointing out the possibility of this type of topological ultranodal state in the Fe-based materials may shed light on the intriguing phenomenology of the iron superconductors that cannot be otherwise explained by conventional means, and encourage experimentalists in their search for new systems with Bogoliubov Fermi surfaces.

In the current ms., we generalized the ideas proposed by Agterberg, Brydon, Timm and Schnyder to a multiband case of spin 1/2 electron pairs, models that describe a class of materials quite different than those treated in Refs. 13 and 14. In addition, by minimization of the Pfaffian with respect to the band energies, our work identifies and examines a previously unexplored topological transition between fully gapped superconducting state to the gapless ultranodal state with residual DOS. Crucially, our work ties together a number of disparate and often puzzling empirical observations in FeSe,S, including: 1) the existence of residual tunneling DOS at zero frequency in the superconducting state of FeSe,S as observed by STM, residual C_v/T at limiting zero temperature, and their respective abrupt jumps together with κ/T at $x = 0.17$; 2) the transition to these phases from a superconductor with more conventional properties; 3) the monotonic decrease of intra-pocket gaps as observed by quantum oscillations and ARPES; 4) the proximity of the nematic quantum critical point to the creation of residual density of states; 5) the existence of strong SOC in FeSe,S; 6) finally, the relatively small value of inter-pocket pairing needed to transition into the “SC2” phase (ultranodal) as compared to the intra- pocket pair.

This last point is worth commenting on further in the historical Fe-based context. From a symmetry point of view, many authors have proposed interband or interorbital pair states since the beginning of the field. But while allowed by symmetry, they are suppressed dynamically because they inevitably involve pairing electrons away from the Fermi level, with concomitant loss of condensation energy. On these grounds, it is expected that if there is any admixture of interband pairs in the pair wave function, it should be small. Naively, such states should therefore not be expected to play an important role in Fe-based physics compared to intraband pairs. On the other hand, we describe here an

explicit situation where a small interband pair can dominate the physics if the intraband pair wave function has a node or deep minimum in exactly the right place, which seems remarkably to occur in the S-doped FeSe system. This is therefore not only quite a novel implementation of the general notion of a topologically protected Bogoliubov Fermi surface, but also one which also may be expected to occur elsewhere since such large gap anisotropies are observed in many Fe chalcogenide systems. Furthermore, our results point to a key prediction for which evidence has been reported recently from experiments in the parent compound - the existence of TRSB in the highly doped FeSe,S superconducting phase.

Response to Reviewer #1

The reviewed article "Topological Ultranodal pair states in iron-based superconductors" by Setty et al. proposes an exotic new pairing state near the nematic transition in the iron selenides. This is a time-reversal symmetry-breaking (TRSB) pairing state with interband pairing, which leads to the formation of Bogoliubov Fermi surfaces where point nodes are expected. This results in low-temperature thermodynamic behaviour closer to a Fermi liquid as opposed to a nodal superconductor, which the authors call "ultranodal". This scenario can explain the abrupt increase in zero-energy spectral weight observed upon doping FeSe as reported in references 15 and 16.

Response: The referee's summary of our work is accurate and we are happy to hear his/her opinion that our proposal can potentially explain the experimental evidence from the mentioned references.

The manuscript is thought-provoking. However, I don't think that the manuscript warrants publication in Nature Communications. The main novelty here is the application to the pnictides, but technically the authors don't seem to significantly go beyond the analysis of refs. 13 and 14. More interesting questions are left unaddressed: how could such a TRSB state arise? Considering the large residual density of states, is such a superconducting phase stable?

Response: We are encouraged to know that the referee considers our work to be thought-provoking. We are also hopeful that our discussion at the beginning of the reply will address all the concerns and criticisms raised by the referee regarding the novelty of our work and the question of the time-reversal symmetry breaking.

While the notion of an extended Bogoliubov surface which explicitly breaks band-space T -symmetry (without recognition of its topological character) goes back at least to the early days of the cuprates (e.g. loop current order (cf. Wang and Vafeek, Phys. Rev. B 88, 024506 (2013)), we believe that the first demonstration of the topological transition into such a TRSB, CP protected topological phase in a spin-1/2 system is novel. Hence, we hope that publication of our results in Nature Communications can provide the necessary visibility to the proposed gapless topological phase as well as its ability to resolve multiple experimental conundrums in iron superconductors.

Let us address the question about stability. We were somewhat uncertain how to interpret the referee's remarks here. Obviously the existence of a large residual DOS in the SC state does not necessarily imply any instability of the superconductor, as seen, e.g., from the example of classic Abrikosov-Gor'kov gapless superconductivity with paramagnetic impurities just before T_c vanishes. The original work of Agterberg and coworkers studied the stability of Bogoliubov surfaces. A free energy analysis shows that inter-band pairing acts like a pseudo internal magnetic field. Hence, the stability of a superconducting state with extended zero energy surfaces can be understood in terms of how an external B field recovers the normal state DOS with increasing magnitude of the field.

Moreover, I found the manuscript to be a bit sloppy. Specifically:

1. The authors introduce the theory for a two-band model, but then immediately switch to a three-band model, thus rendering the Pauli matrix notation of the introduction confusing.

Response: We thank the referee for pointing this out. Essentially, while we work primarily with a three-band model to compare as directly as possible to experiment, for pedagogical purposes, the explanation of the topological character of the state can be given more easily with a two-band model. In the new version of the manuscript, we have explicitly clarified the change in matrix notation while discussing the three band case.

2. The meaning of fig. 1 is unclear: do the insets show the form of the gap used in their simulations? If so, where do the Bogoliubov Fermi surfaces appear? Also, what is the new pocket appearing in the $x=0.22$ sketch?

Response: The insets in Fig. 1 are sketches of the intra-pocket gaps as deduced from experimental evidences and qualitatively resemble the gaps employed in our calculation. This is discussed in the caption of Fig 1. The exact location of the Bogoliubov surface with respect to the original normal state Fermi surface is shown in Fig 2 for the parameters used. As seen, the Bogoliubov surfaces almost (but not entirely) overlap with the normal state Fermi surface. The bands, however, are split according to the relative magnitudes of the inter- and intra-band pairings and TRSB components. The new central hole pocket appearing in the $x = 0.22$ sketch is known from ARPES to arise after a Lifshitz transition. Its presence and the value of the gap on it are irrelevant to our analysis, since only one

pair of gaps need satisfy the condition $\Delta_{\text{intra}} < \delta$ to create the topological state.

3. I don't think that the term "ultranodal" is helpful. The low temperature behaviour of the heat capacity is clear from the term "Bogoliubov Fermi surface".

Response: When we examined different terminologies used in prior literature on this subject, we realized that the phrase "Bogoliubov Fermi surface" was used to describe the zero energy contours in the superconducting state (which can arise for a multitude of reasons), and we did not find any suitable nomenclature for the resulting superconducting state itself. Hence we thought it might be appropriate to use the term "ultranodal state" to describe the resulting superconducting state with extended Bogoliubov surfaces, since the density of quasiparticle excitations is qualitatively larger than any gap nodal system.

4. Is plot fig 2e supposed to be the density of states or the tunneling conductance? The former terminology is used in the text, whereas the latter is used for the figure. It seems to me that a calculation of the density of states would be simpler and much less noisy.

Response: The plot in Fig 2e is the tunneling conductance, in the sense that the density of states has been convolved with a Fermi function (cf. J. Hoffman, Rep. Prog. Phys. 74 (2011) 124513). We have modified the text and used the term "tunneling conductance" instead of DOS to describe Fig 2e.

5. The authors appear to mischaracterize refs. 13 and 14 when they state that "Time reversal symmetry can also be broken when spin-orbit coupling generates a pseudo-magnetic field differentiating between up and down spins." The spin-orbit coupling in refs. 13 and 14 is not essential for the TRSB, nor for the appearance of Bogoliubov Fermi surfaces.

Response: We thank the referee for pointing this out. We have modified the text in the introduction to reflect this comment of the referee.

Response to Reviewer #2

The authors considered a phenomenological model for superconductivity in FeSeS, which is shown in experiments to have a finite ratio of the Sommerfeld coefficient in specific heat in the superconducting state, implying the existence of gapless excitations. In the theoretical model, intra- and inter-pocket pairings, singlet and triplet, and time-reversal symmetry breaking, are included in such a way that the BdG quasiparticles have a Fermi surface, and the superconducting state is dubbed ultranodal. This naturally explain the behavior of the puzzling specific heat. The authors believed that the transition from fully gapped state to the ultranodal state is topological in nature, as reflected in the change of the sign of the Pfaffian of the matrix Hamiltonian.

Response: We thank the referee for summarizing our work and are delighted to hear that the referee thinks we are able to address the question of the "puzzling specific heat" and provide a natural explanation for it.

While the proposal sounds interesting, I have some comments for the authors to consider before I could make a decision on the publication of the manuscript.

Response: We are glad to hear that the referee considers our work to be interesting. We are hopeful that our responses below to the referee's questions will help her/him make a final decision in favor of our manuscript.

[1] The ultranodal state requires fine tuning of the parameters, and as mentioned in the text, in some cases, the transition is even sensitive to some infinitesimal tuning of inter-pocket coupling. The need of fine tuning of the various pairing parameters, and the lack of adiabatic continuity in the inter-pocket coupling, make the theory less compelling.

Response: We want to emphasize that since the ultranodal state is topologically protected by CP and TRSB, by definition, the state is *robust* to fine tuning of Hamiltonian parameters. In particular, for the model presented in our manuscript, deep in the topological phase, as long the condition for the Pfaffian sign change is satisfied, any fine tuning of the hopping and/or intra-/inter-pocket gaps will not affect the character of the topological state.

In the specific comment of the referee, we suspect he/she is referring to the text "Similar tuning of the Bogoliubov Fermi surface locations can be achieved with inter-band hoppings". By this statement we only mean that one can obtain the condition for the Pfaffian sign change either by increasing the inter-pocket order parameter or reducing the intra-pocket order parameter. Deep in the topological phase – achieved either of the two ways – the existence of Bogoliubov surfaces is immune to fine tuning of model parameters. On the other hand, the exact *location* of the extended surfaces in the Brillouin zone is not protected by symmetries and hence can depend on model parameters. We have tried to clarify the language used to describe the BFS locations to avoid further confusion. Finally, it is possible that the referee is referring to our very specific choices for certain Hamiltonian parameters. Creating the conditions for the topological phase to exist requires some carefully chosen constraints on, e.g. the intraband gaps; but for each of these choices, we have justification from experiment.

Close to the topological transition, the system becomes sensitive to parameters in the same sense as all other well established topological phases such as QSHE and topological insulators/superconductors. In the latter systems, close the critical point where the band dispersions close and reopen, small changes in the hopping and/or pairing can push the electronic structure either into the topological or trivial phase.

[2] The need of time-reversal symmetry breaking is also a concern. Is there any evidence of this symmetry breaking in experiments?

Response: As this question was also raised by the first referee, we have pointed out experimental evidence for the TRSB this at the beginning of our reply.

[3] The Sommerfeld coefficient in experiments indicate, at most, the existence of low lying quasiparticles. But the model here implies more than that. For example, it actually implies the existence of dispersive low lying quasiparticles, and this should imply the existence of such states in transport, e.g., thermal conductivity. Is there any evidence in this respect?

Response: As correctly inferred by the referee, the presence of dispersive bands close to E_f (i.e., with non-zero v_f) would mean that thermal conductivity should also see a large jump close to zero T . Indeed, measurements of the Tokyo-Kyoto group do show a large increase in residual $\kappa(T \rightarrow 0)$ in the topological phase. This supports our claim of an ultranodal state for large doping (see Sato et al. PNAS **115**, 1227-1231 (2018)).

Traditionally, the existence of a finite residual κ/T as $T \rightarrow 0$ is taken as an indication that the unconventional superconductor in question has line nodes, is a signature of quasi-universal transport, and its value does *not* scale with disorder. It is expected that the S dopants, away from the Fe plane, provide relatively weak scattering anyway. This helps explain why for low x , the FeSe,S material displays a small thermal conductivity that stays roughly constant over several low dopings. On the other hand, the $x=0.22$ doping residual κ/T jumps significantly. In our picture, this is entirely consistent with the system having entered the ultranodal state, where an additional set of normal-metal like quasiparticles from the Bogoliubov Fermi surface can carry a thermal current.

[4] The key of the phenomenological theory is the multiorbital physics that enables the spin-singlet and spin-triplet to be both even and odd under inversion, since the orbitals can make up the antisymmetry. In this respect I would like to draw the authors' attention to the earliest paper on possible p-wave spin-singlet orbital-triplet pairing scenario for iron pnictides, Phys. Rev. Lett., 101 (2008) 057008. Without recognition, that state also has a BdG Fermi surface.

Response: We thank the referee for drawing this reference to our attention. We would like to emphasize that for an ultranodal state, one requires the Pfaffian to be well defined. This can be obtained by a combined CP as well as broken T -symmetry. Indeed, there are several works in early literature on high T_c superconductivity (e.g., loop current order in the cuprates) where these symmetries were satisfied but not recognized at that time. In the reference pointed by the referee, we do not quite see how these symmetry conditions are satisfied while being consistent with the Pauli principle.

[5] As the authors mentioned, the interpocket pairing does not enjoy the infrared divergence in the pairing susceptibility. It is here that the authors argued that an infinitesimal interpocket coupling would rescue the logarithmic infrared divergence. While this is indeed correct because pocket hybridization enables the contents of the Bloch state to redistribute, adiabatic continuity implies that such a divergence must be proportional to the fourth power of the infinitesimal coupling. It is unclear to me why such a weak effect should be able to drive superconductivity and be more favorable than intrapocket pairing. Without a mechanism to stabilize the model under concern, the theory is purely phenomenological, and is not exclusive, since other phenomenological fitting would work as well.

Response: For our proposal to work, we do not require the inter-pocket pairing to be more favorable than intrapocket pairing. On the contrary, the scenario we present for FeSe,S has large intrapocket gaps and small interpocket ones, much more physically reasonable. We point out how the intrapocket gap nodal structure in such a situation allows the condition for the creation of Bogoliubov Fermi surface to be fulfilled for a small subset of k-vectors. All we need is that the intra-pocket gaps go to zero at *at least* one point on the Fermi surface for the condition of Pfaffian sign change to occur (see Eq 2). This generally holds true for repulsive interactions and can already give rise to a finite extended "ultranode" close to the original point node of the intra-pocket pair. The requirement is very easily satisfied given that many unconventional superconductors have nodal intra-pocket pairing or deep gap minima.

[6] Last and most importantly, given the existence of the BdG Fermi surface and hence a finite density of propagating zero-energy BdG quasiparticles (which now carry charge since they are no longer symmetric combinations of particles and holes), the authors should check the stability of the superconducting state to phase twisting, namely, the superfluid density. It is known, at least in one example, that the existence of the BdG Fermi surface leads to zero or even negative superfluid density. See EPL, **85** (2009) 57007.

The referee is right, and we are grateful for the remark. The existence of a Bogoliubov surface can be associated with zero and/or negative superfluid density. We have calculated the superfluid density for our model, and in the new ms. comment on the stability of the topological state, and present results for the superfluid density in various special cases in a new section of the supplementary material. To summarize our findings qualitatively: for any model consisting entirely of intraband pairing, the $T \rightarrow 0$ superfluid density is identical to the normal state fermion density,

as expected from BCS mean field theory. In the presence of interband pairing, the superfluid density is suppressed, and in the limit where interband pairing dominates, can become negative (unphysical). This tendency is exacerbated to the extent that the mass difference $m_1 - m_2$ for the two bands becomes large. However, *in the limit considered in the paper and applied to FeSe,S, where the maximum intraband gaps are larger than their interband counterparts, the superfluid density is always positive.* This investigation suggested by the referee's remark, while enlightening, has therefore not altered our conclusions.

A discussion of the issue of superfluid density now appears in the main manuscript with details moved to the Supplemental Material.

To summarize, the paper in the present form lacks the scientific rigor for publication in NC.

We trust that with the detailed responses above we have assuaged the referee's concerns about scientific rigor.

Response to Reviewer #3

In this manuscript, the authors propose an interesting superconducting state of $\text{FeSe}_{1-x}\text{S}_x$ in the tetragonal phase ($x > x_c=0.17$). Recent STM study [15] reported that the superconducting gap structure becomes suddenly isotropic for $x > x_c$, and the residual DOS increases with x . This experiment attracts great attention because it indicates the intimate relation between superconductivity and nematic order. To explain nontrivial superconducting DOS of $\text{FeSe}_{1-x}\text{S}_x$ for $x > x_c$, the present authors applied the idea of superconductivity with Bogoliubov Fermi surface, which appears when the time-reversal symmetry (TRS) breaks, and the inter-band Cooper pairs dominate over the intra-band ones. The obtained DOS in Fig.4 seems to explain the result of STM study in Ref. [15].

Response: The referee's summary of our work and context of our results is accurate, and we thank him/her for the careful reading of the manuscript.

I think that the exotic superconductivity proposed in this manuscript is very interesting, so the present study will stimulate many experimentalists in this field. On the other hand, the authors do not discuss possible origin of this TRS breaking inter-band pairing state. Actually, one has to assume very large inter-band pairing interaction for $x > x_c$, much larger than the intra-band pairing interaction for $x < x_c$. However, this compound becomes weak-coupling with increasing x . To explain monotonous increase of residual DOS with $x(> x_c)$, the inter-band pairing interaction should remain large away from the nematic critical point $x = x_c$.

Response: We are glad to know that the referee considers our work interesting, and we agree with the referee that the results could stimulate experimental activity in this direction. With regards to TRSB and inter-band pairing, this is an issue that was also raised by the two other referees and is discussed at the beginning of our reply.

:

For our proposal to work we do *not* require the inter-pocket pairing to be more favorable than intra-pocket pairing. On the contrary, the scenario we present for $\text{FeSe}_x\text{S}_{1-x}$ has large intrapocket gaps and small interpocket ones, much more physically reasonable. We point out how the intrapocket gap nodal structure in such a situation allows the condition for the creation of Bogoliubov Fermi surface to be fulfilled for a small subset of k -vectors. All we need is that the intra-pocket gaps go to zero at *at least* one point on the Fermi surface for the condition of Pfaffian sign change to occur (see Eq 2). This generally holds true for repulsive interactions and can already give rise to a finite extended "ultranode" close to the original point node of the intra-pocket pair. The requirement is very easily satisfied given that many unconventional superconductors have nodal intra-pocket pairing or deep gap minima.

The Bogoliubov Fermi surface (shown by blue lines in Fig.2) appears at the momentum k , in the case that two bands with $E_1(k)$ and $E_2(k)$ are very close to the Fermi level. For this reason, the band structure near the Fermi level is very important information. Therefore, I believe that the authors should show the model band structure, and make comparison with experimental band structure in (for instance) Phys. Rev. B **96**, 121103 (2017) in detail.

Response: We are not entirely sure we understand the referee's desires here. He/she refers to the two-pocket model with bands E_1 and E_2 , but we explicitly compare a three-pocket model with the experimental data. In case the distinction between two and three pockets is the source of the confusion, we apologize. The mathematical structure of the model was easier to discuss pedagogically for the two band case, which we presented first, but all "real" calculations were performed with the three band Hamiltonian. We have added language to correct the confusion. As for the exact nature of the electronic structure used, our band structure is fairly far from the realistic case as measured, e.g. by ARPES. We say explicitly that we have used parabolic bands, giving the Fermi surface shown in Fig. 2; in addition, we have not paid particular attention to making sure that the correct Fermi velocities are reproduced. But these details are really irrelevant to our conclusions, since we are primarily interested in the qualitative aspects of the topologically protected state. Making the details more realistic will move the topological transition and change the size and location of the Bogoliubov Fermi surface, but not the existence of the topological state displaying these surfaces.

In summary, the proposed exotic superconductivity with Bogoliubov Fermi surface for $FeSe_{1-x}S_x$ is very interesting. The present will promote many experimental studies to detect the Bogoliubov Fermi surface and to find the evidence of the TRS breaking in the superconducting state. On the other hand, the present authors do not discuss possible origin of the inter-band pairing interaction based on the characteristic band structure of $FeSe_{1-x}S_x$ near E_F . The idea of Bogoliubov Fermi surface was originally discussed in Ref. [13] in a general model. An alternative explanation for the residual DOS, which increases with x monotonically, might be given by dirty d-wave superconductivity. Therefore, I feel a hesitation to recommend the present manuscript for publication in Nature Communications.

Response: We would like to comment on the referee's concern that an alternative explanation may be made invoking the dirty d-wave theory. We believe this scenario to be ruled out for the following reasons: first, the samples seem relatively clean as seen in STM (one of the authors discussed this question explicitly with T. Hanaguri, who made the measurements, who says he sees much too little disorder to be consistent with the large residual specific heat values). Secondly, this statement of the high purity of the system is consistent with the fact that quantum oscillations experiments observe sharp, well-defined frequencies. Finally, the dopants in this system are located away from the Fe planes, thus reducing their effects on the superconducting state. They are not expected to be strong scatterers capable of inducing a significant residual DOS.

Furthermore, the jump in the residual DOS at $x > x_c$ is abrupt, as stated explicitly in the title of the experimental paper. One would imagine that if the residual DOS effects are related to impurities, there should be a gradual turn-on of the zero temperature Sommerfeld coefficient and zero frequency DOS even for $x < x_c$. This is however inconsistent with experiment.

The referee in his summary points out that we are not the first to propose the idea of a Bogoliubov Fermi surface (nor, in fact, was Ref. 13. The idea has been put forward in a number of contexts, including loop current order in cuprate superconductors), suggesting that part of his/her hesitation may be related to the question of novelty. We would like to refer to the following brief discussion at the top of our reply of why we think our results represent a very significant step forward beyond the work of Ref. 13.

Reviewers' comments:

Reviewer #1 (Remarks to the Author):

The authors have resubmitted their article "Topological Ultranodal pair states in iron-based superconductors" with changes suggested by the referees. In particular, the authors have improved the discussion and also performed additional calculations to show the stability of the system.

My belief is that this article is a somewhat marginal case. On the one hand, the paper represents an interesting application of the concept of Bogoliubov Fermi surfaces to a material of current high interest, presents important experimental signatures of such a state, and seems to be technically correct. On the other hand, the motivation for their work remains deficient, the discussion is difficult to follow in places, and the paper makes a number of misleading statements.

Nevertheless, with some work these issues can be overcome and the paper may be suitable for publication in Nature Communications. Below I consider in detail each point:

Motivation:

* As pointed out by a number of referees, the time-reversal symmetry-breaking state proposed by the authors is a rather radical solution to the problem posed by the measurements on Fe(Se,S). In their response letter, the authors cite recent conference talks where hints of such a state have been found. This lends much more support to their physical picture. Since the abstracts of these talks are on the public record, I suggest that the authors reference them in their manuscript.

Discussion:

* The discussion in the section "Model" remains dense and potentially confusing for a general audience. Part of the problem is that so many of the equations are in-line. Moreover, notations are used unevenly: does a "hat" indicate a matrix or an operator? Is it necessary to introduce the "underlined" notation?

* The underlined pairing potential is problematic, since the intraband pairing potential (Δ_s) is almost immediately modified so that it has different values on the various pockets. In the notation of the authors, this appears to correspond to a pairing potential with matrix structure $\tau_z \otimes i\sigma_y$. This should be included in the full expression for the pairing potential.

* I would also like to have a better understanding of the pairing potential that the authors are proposing. What are the symmetry properties of the interband pairing? Is the resulting TRSB state something like a $d+is$ state? Is it plausible for this material?

* The Pfaffian, although crucial to the author's argument, is introduced with the extremely unclear caveat that "inter-band couplings are set to zero" - what does this mean and how would it change the final result if it is relaxed?

I recommend a thorough overhaul of this section with an eye to making it as self-contained and accessible as possible.

Misleading statements:

* The authors have not corrected their statement that "Time-reversal symmetry can also be broken

when spin-orbit coupling generates a pseudo-magnetic field differentiating between up and down spins [13,14]." Apart from the fact that the concept of a "pseudo-magnetic field" is not explained, this is a misunderstanding of references 13 and 14: in these papers, the pairing potential itself breaks time-reversal symmetry; the pseudo-magnetic field arises in an effective low-energy description upon treating the interband pairing perturbatively.

This confusion appears elsewhere with "it is broken here because the inter-orbital pairing acts as a pseudo-magnetic field". Moreover, this is notably difficult to understand since the authors have abruptly slipped from a band to an orbital picture.

* I continue to dislike the concept of "ultranodal" pairing: "surface nodal" seems like the obvious generalization of "point nodal" and "line nodal". Ultranodal could confuse readers about the relation to Bogoliubov Fermi surfaces.

Reviewer #3 (Remarks to the Author):

I read the revised manuscript with care, and found that the authors improved the manuscript by replying referees' comments. At present, the ultranodal SC state may not be unique solution of the residual DOS in Fe(Se,S), because competition between spin and orbital fluctuations and/or strong orbital selectivity cause wide variety of SC gap structure in Fe-based superconductors. However, the predicted TRS breaking and the emergence of interesting Bogoliubov Fermi surface will stimulate experimental efforts, such as ARPES and STM/STS studies. Therefore, I recommend the manuscript for publication if the authors respond to the following brief comment: The mechanism of superconductivity or the pairing interaction V_{inter} and V_{intra} for the ultranodal SC state is very nontrivial. Let us consider the BCS equation: For simplicity, the right-hand-side of BCS equation for intra-band gap is $\sim D(0)V_{\text{intra}} \log(\omega_c/T_c) \Delta_{\text{intra}}$, whereas that for inter-band gap is $\sim D(0)V_{\text{inter}} \log(\omega_c/|E_1-E_2|) \Delta_{\text{inter}}$, where ω_c is the cutoff energy, $D(0)$ is the DOS in the normal state, $|E_1-E_2|$ is the band-splitting energy. When $|E_1-E_2| \gg T_c$, very large V_{intra} will be necessary for the relation $\Delta_{\text{inter}} \sim \Delta_{\text{intra}}$ that is a necessary condition for the ultranodal SC. For this reason, I made the following comment in the second paragraph of my first referee comment: "The Bogoliubov Fermi surface (shown by blue lines in Fig.2) appears at the momentum k , in the case that two bands with $E_1(k)$ and $E_2(k)$ are very close to the Fermi level. For this reason, the band structure near the Fermi level is very important information." Therefore, I advise the authors to add a short discussion on the mechanism of superconductivity or the pairing interaction to the manuscript.

Typo: In the right column on page 5, $E_{\{-}}$ should be $+0.6$.

Response to Referees:

Second response to Reviewer #1

The authors have resubmitted their article "Topological Ultranodal pair states in iron-based superconductors" with changes suggested by the referees. In particular, the authors have improved the discussion and also performed additional calculations to show the stability of the system.

My belief is that this article is a somewhat marginal case. On the one hand, the paper represents an interesting application of the concept of Bogoliubov Fermi surfaces to a material of current high interest, presents important experimental signatures of such a state, and seems to be technically correct. On the other hand, the motivation for their work remains deficient, the discussion is difficult to follow in places, and the paper makes a number of misleading statements. Nevertheless, with some work these issues can be overcome and the paper may be suitable for publication in Nature Communications. Below I consider in detail each point:

Response: We thank the referee for his/her opinion about our work and we are encouraged to hear that our paper may be suitable for publication in Nature Communications. In our response below, we address the issues raised by the referee regarding the motivation and other statements of concern. In the new version of the manuscript, we have incorporate these changes suggested by the referee and also simplify the discussion and notation of the model to improve readability.

Motivation:

* As pointed out by a number of referees, the time-reversal symmetry-breaking state proposed by the authors is a rather radical solution to the problem posed by the measurements on Fe(Se,S). In their response letter, the authors cite recent conference talks where hints of such a state have been found. This lends much more support to their physical picture. Since the abstracts of these talks are on the public record, I suggest that the authors reference them in their manuscript.

Response: As suggested by the referee, we have included a reference to these talks in the introduction of the revised manuscript.

Discussion:

* The discussion in the section "Model" remains dense and potentially confusing for a general audience. Part of the problem is that so many of the equations are in-line. Moreover, notations are used unevenly: does a "hat" indicate a matrix or an operator? is it necessary to introduce the "underlined" notation?

Response: As suggested by the referee, we have tried to improve readability for a general audience by removing the in-line equations and spacing out the text in the revised manuscript. In the entire manuscript, the use of hats is now confined to operators and the equations have been simplified by removal of underlined notation.

* The underlined pairing potential is problematic, since the intraband pairing potential (Δ_s) is almost immediately modified so that it has different values on the various pockets. In the notation of the authors, this appears to correspond to a pairing potential with matrix structure $\tau_z \otimes i\sigma_y$.

Response: We agree with the referee. The underlined pair potential matrix we wrote in page two (second column) of the original manuscript holds only when the gaps on both pockets are the same. When this is not the case, one would require a matrix form $\tau_z \otimes i\sigma_y$. For simplicity and to remain within word limit, we have removed matrix notation for the pairing Hamiltonian and stick with operator notation. The intrapocket pairing is simply labeled $\Delta_i(\mathbf{k})$ from now on.

* I would also like to have a better understanding of the pairing potential that the authors are proposing. What are the symmetry properties of the interband pairing? Is the resulting TRSB state something like a d+is state? Is it plausible for this material?

Response: In the paragraph below the definition of \hat{H}_Δ in the new version of ms, we describe the symmetry properties of the interband terms. All three terms in the pairing potential respect C and P symmetries. However, while the terms Δ_0 and $\Delta_i(\mathbf{k})$ maintain T symmetry, δ breaks it. As outlined in the Discussion section, a pure TRSB

state such as $d + is$ cannot give rise to Bogoliubov Fermi surfaces in our model. We rather require TRSB in spin to achieve Bogoliubov Fermi surfaces.

* The Pfaffian, although crucial to the author's argument, is introduced with the extremely unclear caveat that "inter-band couplings are set to zero" - what does this mean and how would it change the final result if it is relaxed? I recommend a thorough overhaul of this section with an eye to making it as self-contained and accessible as possible.

Response: In a previous version of the manuscript we used the term "inter-band couplings" to reflect off-diagonal terms in normal state Hamiltonian. However, as we are working in band-space throughout the current manuscript, these terms are automatically zero. Hence, the phrase "inter-band couplings are set to zero" is redundant and we have removed it in the new version of the paper. We apologize for this confusion, and thank the referee for pointing this out.

Misleading statements:

* The authors have not corrected their statement that "Time-reversal symmetry can also be broken when spin-orbit coupling generates a pseudo-magnetic field differentiating between up and down spins [13,14]." Apart from the fact that the concept of a "pseudo-magnetic field" is not explained, this is a misunderstanding of references 13 and 14: in these papers, the pairing potential itself breaks time-reversal symmetry; the pseudo-magnetic field arises in an effective low-energy description upon treating the interband pairing perturbatively.

Response: It looks like we may have missed correcting this statement in our previous revised version of the manuscript. We thank the referee for pointing it out again and offer our apologies. The referee is right that "spin-orbit coupling in refs. 13 and 14 is not essential for the TRSB, nor for the appearance of Bogoliubov Fermi surfaces". To be consistent with the paper of Agterberg et. al, we have modified this to "Time reversal symmetry can also be broken in the presence of a pseudo-magnetic field arising from interband pairing [1, 2]. It has been pointed out that such terms can be generated by spin-orbit coupling in iron based systems [3, 4]". The references appear below. These new statements are consistent with the original PRL of Agterberg, Brydon and Timm – "We further interpret these Fermi surfaces in terms of a pseudomagnetic field arising from interband Cooper pairs, here referred to as "interband pairing". "

This confusion appears elsewhere with "it is broken here because the inter-orbital pairing acts as a pseudo-magnetic field". Moreover, this is notably difficult to understand since the authors have abruptly slipped from a band to an orbital picture.

Response: Like the point mentioned above, we attribute the essence of the phrase which the referee points out to the original PRL of Agterberg, Brydon and Timm – "We further interpret these Fermi surfaces in terms of a pseudomagnetic field arising from interband Cooper pairs, here referred to as "interband pairing". " In the new manuscript, we have modified our sentence to "it is broken here because the inter-pocket pairing gives rise to a pseudo-magnetic field". We believe this phrase is consistent with the previous work of Agterberg and coworkers.

* I continue to dislike the concept of "ultranodal" pairing: "surface nodal" seems like the obvious generalization of "point nodal" and "line nodal". Ultranasodal could confuse readers about the relation to Bogoliubov Fermi surfaces

Response: While writing the paper, we were looking for a simple description of the superconducting state resulting from Bogoliubov Fermi surfaces as there was none existing in current literature. We wanted a terminology that is both accurate and distinguishable from a trivial superconducting state, but at the same time phonetically easy to refer to during discussions. We think "surface nodal superconducting state" or "extended nodal superconducting state" are also very appropriate names but can be verbally lengthy especially while being referred to multiple times.

Second response to Reviewer #3

I read the revised manuscript with care, and found that the authors improved the manuscript by replying referees comments. At present, the ultranodal SC state may not be unique solution of the residual DOS in Fe(Se,S), because competition between spin and orbital fluctuations and/or strong orbital selectivity cause wide variety of SC gap structure in Fe-based superconductors. However, the predicted TRS breaking and the emergence of interesting Bogoliubov Fermi surface will stimulate experimental efforts, such as ARPES and STM/STS studies. Therefore, I recommend the manuscript for publication if the authors respond to the following brief comment:

Response: We are encouraged and thankful to the referee for recommending our manuscript for publication.

The mechanism of superconductivity or the pairing interaction V_{inter} and V_{intra} for the ultranodal SC state is very nontrivial. Let us consider the BCS equation: For simplicity, the right-hand-side of BCS equation for intra-band gap is $D(0)V_{intra} * \log(\omega_c/T_c) * \Delta_{intra}$, whereas that for inter-band gap is $D(0)V_{inter} * \log(\omega_c/|E_1 - E_2|) * \Delta_{inter}$, where ω_c is the cutoff energy, $D(0)$ is the DOS in the normal state, $|E_1 - E_2|$ is the band-splitting energy. When $|E_1 - E_2| \gg T_c$, very large V_{intra} will be necessary for the relation $\Delta_{inter} \sim \Delta_{intra}$ that is a necessary condition for the ultranodal SC. For this reason, I made the following comment in the second paragraph of my first referee comment: The Bogoliubov Fermi surface (shown by blue lines in Fig.2) appears at the momentum \mathbf{k} , in the case that two bands with $E_1(k)$ and $E_2(k)$ are very close to the Fermi level. For this reason, the band structure near the Fermi level is very important information. Therefore, I advise the authors to add a short discussion on the mechanism of superconductivity or the pairing interaction to the manuscript.

We believe that the referee is referring to the following critique. In general, we are appealing to an interband pairing term with momenta \mathbf{k} and $-\mathbf{k}$, such that if one electron is at the Fermi surface, the other is perforce at an energy $E_g = E_1 - E_2$ away from the Fermi surface. Even though this energy may be within a pairing cutoff ω_c (“Debye energy”), such that the BCS interaction is attractive, the finite energy splitting E_g leads to a suppression of T_c and Δ_{inter} . Δ_{inter} is suppressed by a factor $\log|\omega_c/E_g|$ if $E_g \gg T_c$. To promote the topological transition to a Bogoliubov Fermi surface with constant interactions, one would indeed need $\Delta_{intra} \sim \Delta_{inter}$, as the referee says. The referee appears to take this as a sign that the band structure near the Fermi level should be rather unusual, perhaps displaying some near-degeneracies of different bands in order to minimize E_g . We hope we have understood properly the comment of the referee in this regard.

While the referee’s concern is indeed warranted in the simple case with constant interactions, we are proposing a novel way around this problem, such that such fine tuning of the band structure near the Fermi level is not required. This relies on the empirically known nodal or near-nodal structure of the principal intraband gaps. Thus we do *not* require that $\Delta_{intra}(\mathbf{k}) \sim \Delta_{inter}$ for all momenta, but only over a small range of \mathbf{k} near the nodes. The relationship between the intra- and interband interactions can in fact be arbitrary provided the intraband interaction is highly anisotropic in \mathbf{k} . We have assumed here that V_{intra} is generically large but strongly momentum-dependent, consistent with experiment, and that V_{inter} is small, as expected for an interaction induced by weak spin-orbit coupling.

We agree that this situation should be clarified and we have added several sentences to the manuscript in this context.

Typo: In the right column on page 5, E_- should be $+0.6$.

Response: We thank the referee for pointing this out. The revised manuscript contains the corrected values.

-
- [1] D. F. Agterberg, P. M. R. Brydon, and C. Timm, Phys. Rev. Lett. **118**, 127001 (2017).
 - [2] P. M. R. Brydon, D. F. Agterberg, H. Menke, and C. Timm, Phys. Rev. B **98**, 224509 (2018).
 - [3] O. Vafek and A. V. Chubukov, Phys. Rev. Lett. **118**, 087003 (2017).
 - [4] P. M. Eugenio and O. Vafek, Phys. Rev. B **98**, 014503 (2018).